# Endemic vascular plants provide reliable indicators for mapping seasonally dry tropical forests

Mayra Flores-Tolentino[1], José Luis Villaseñor[2], Enrique Ortíz[2], David A. Prieto-Torres[3], Guillermo Ibarra-Manríquez[1]*

1 Instituto de Investigaciones en Ecosistemas y Sustentabilidad, Universidad Nacional Autónoma de México, Morelia, Michoacán, Mexico, 2 Instituto de Biología, Departamento de Botánica, Universidad Nacional Autónoma de México, Ciudad de México, Mexico, 3 Laboratorio de Biodiversidad y Cambio Global (LABIOCG), Facultad de Estudios Superiores Iztacala, Universidad Nacional Autónoma de México, Estado de México, Mexico

* gibarra@iies.unam.mx

## Abstract

Plant communities are unevenly distributed in space, shaped by both abiotic and biotic factors. Several methods have been developed for delineating their extent, including the spatial analysis of vegetation patterns using tools such as vegetation maps, climate-based simulations, and the use of characteristic species distribution. However, limited knowledge exists about which species are most suitable for this purpose. In this study, we aimed to delimit the seasonally dry tropical forest (SDTF) in Mexico based on the distribution area of vascular plant species endemic to Mexico and registered to this biome. Endemic species serve as key indicators for delineating biomes, highlighting regions with stable conditions and unique evolutionary and biological characteristics. The occurrence records of species were obtained from the Global Biodiversity Information Facility (GBIF) database, and ecological niche models were generated using the ENMTML package in R. The boundaries of the SDTF were delineated by stacking species distribution models, grouping endemic species according to the proportion of their occurrence records located within the SDTF: *i*) ≥50% of records (SDTF 50%), *ii*) ≥75% (SDTF 75%), and *iii*) 100% (SDTF 100%). Model performance was evaluated using Kappa, sensitivity, and specificity metrics. We validated our results using Asteraceae points distributed across Mexico's major biomes and analyzed confusion matrices. A total of 3,673 endemic species were registered, and 228 met the criteria for species distribution modeling. Of these, 96% yielded models with high predictive accuracy. Among the three approaches, the model based on high-affinity species (SDTF 75%) performed best in terms of all evaluation metrics, delineating approximately 14% of Mexico's surface as SDTF. In conclusion, high-affinity species serve as reliable indicators for delineating plant communities with well-defined environmental characteristics, facilitating both the

**Data availability statement:** All relevant data are within the manuscript and its Supporting Information files.

**Funding:** This study was supported by the Dirección General de Asuntos del Personal Académico of the Universidad Nacional Autónoma de México (DGAPA-UNAM; PAPIIT in the form of a grant awarded to DAP-T projects IA202822 and IA208824). The specific roles of this author are articulated in the 'author contributions' section. The funders had no role in study design, data collection and analysis, decision to publish, or preparation of the manuscript.

**Competing interests:** The authors have declared that no competing interests exist.

precise delineation of biomes and their application in conservation ecology and in other biomes.

## Introduction

Plant communities are distributed heterogeneously across space, influenced by both abiotic and biotic factors [1–3]. Several approaches have been proposed for its geographical delimitation, including the analysis of spatial vegetation patterns, the distribution of characteristic species, and the use of tools such as vegetation maps and climate model simulations [e.g., 4]. These methods enable us to understand how a region's prevailing environmental characteristics influence species distributions [3], which is crucial for comprehending biodiversity patterns, ecosystem conservation, and natural resource management [5,6]. Additionally, these approaches enable the identification of areas dominated by specific vegetation types. They can reveal how they may change over time due to various factors, such as vegetation dynamics, climate change, or anthropogenic disturbances [3,7,8]. Interactions between plant species and abiotic factors – such as climate, topography, and geology – establish natural boundaries that define species assemblages and community composition [2,3,9]. Therefore, geographic delimitation of plant communities is crucial for understanding ecological processes across spatial scales and identifying the drivers of transition zones between vegetation types or biomes [10]. However, accurately delimiting vegetation types remains a challenging task due to the complexity of ecological interactions and environmental gradients [11–13]. As a result, there is a pressing need for research that applies integrative approaches to biodiversity assessment, leveraging the growing availability of digital databases, ecological modeling tools, and multivariate analytical techniques, including species distribution models (SDMs) [14].

Combining biodiversity databases with quantitative methods has significantly enhanced the ability to assess and delineate plant communities. For example, one such method is the use of stacked species distribution models (s-SDMs), a relatively recent approach that offers insights into spatial patterns of plant diversity and the characterization of specific communities [5,15,16]. Species distribution modeling (SDM) estimates the realized niche of each species independently, and their results can then be combined to observe community-level patterns [17]. In biogeography, the assembly of individual SDMs is particularly useful for identifying areas of species co-occurrence, making them a fundamental resource for recognizing biotic regions [18]. SDMs are also widely employed to analyze the relationship between species occurrences and environmental conditions [19–21], enabling the identification of suitable habitats and revealing large-scale distributional trends [22–25]. Furthermore, the increasing availability of geospatial information contained in digital databases (e.g., the Global Biodiversity Information Facility, GBIF), combined with multivariate analytical tools, has enhanced the resolution and objectivity of biogeographic regionalization efforts [26–29]. These tools have also proven useful in evaluating the ecological affinities of taxa and biomes [30]. Although various methodologies have been proposed

to define geographical boundaries [26,29,31], considerable uncertainty remains regarding which species or groups of species are most appropriate for accurately defining community boundaries. The appropriate selection of these groups is essential not only for a rigorous characterization of biomes and ecosystems but also for anticipating their responses to potential future change scenarios [11,12,30–33].

In Mexico, at least five main biomes are recognized [34]: humid montane forest, temperate forest, seasonally dry tropical forest, humid tropical forest, and xerophilous scrubland. Among them, the seasonally dry tropical forest (hereafter SDTF) is considered one of the most floristically diverse biomes in the world, especially in the Americas [35–37]. This biome is distributed discontinuously across the country [38,39], with three major regions [40]: *i*) the Pacific coast, including the Balsas Depression; *ii*) the northwest of the Yucatan Peninsula; and *iii*) central Veracruz and southern Tamaulipas, in the Gulf of Mexico. These regions differ primarily in their topography, soil characteristics, biogeographic history, vegetation physiognomy, and floristic composition [39,41]. The Mexican SDTF harbors over 6,000 vascular plant species, nearly 45% of which are endemic [34,42]. Additionally, it is characterized by relatively fertile soils, which have led to an increase in the risk to its conservation due to various factors, including changes in land use, fires, the introduction of exotic species, overharvesting of plants, and water extraction [43]. Deforestation rates are alarmingly high, with only 27% of the original forest cover remaining intact by 1990 and an annual deforestation rate of 1.4% [44]. Consequently, it is considered one of the most threatened biomes in Mexico and the World [36,40,45,46].

Different proposals for delimiting the SDTF in Mexico vary in their methodological approaches. Some of them are based on the empirical knowledge of the biome to define its boundaries [e.g., 36, 38]. In contrast, others are based on the distribution of specific taxonomic groups that commonly include broadly distributed species that do not necessarily reflect the macroecological patterns of the biome [e.g., 31], or on the reclassification of vegetation maps [4]. Consequently, we hypothesized that species with high ecological affinity to the biome would provide a more precise delineation of their distribution than widely distributed species. Accordingly, this study aimed to delimit the SDTF in Mexico using the distribution of vascular plant species that are endemic or exclusive to this biome. This strategy has proven more effective for geographically delimiting plant communities than approaches based on forest inventories or multivariate spatial analyses [5,15,17]. To assess which species best define the SDTF boundaries, we grouped the species into three categories according to the proportion of their occurrence records within the biome: *i*) ≥50% of records (SDTF 50%), *ii*) ≥75% (SDTF 75%), and *iii*) 100% (SDTF 100%). This approach allows us to evaluate which set of species provides the most accurate and ecologically meaningful delineation of the SDTF in Mexico [6].

This study is necessary to improve the understanding and conservation of the SDTF in Mexico, one of the most threatened biomes in the Neotropical region. Despite the availability of vegetation maps and climate models, the precise geographic boundaries of this biome remain poorly defined, and existing approaches often fail to capture areas of high ecological importance. By focusing on endemic species with strong ecological fidelity to the SDTF, our study provides a more accurate and ecologically grounded delineation of its boundaries. This refined mapping is crucial for guiding conservation planning, prioritizing protection areas, and understanding the spatial distribution of plant communities at the national scale.

## Materials and methods

### Data cleaning and species selection

The species occurrence records were obtained from the GBIF database [47]. The filters applied during the data download included records with herbarium specimens, restricted to vascular plants within Mexico as the geographic boundary. This search yielded a database with 4,856,783 records corresponding to 72,539 species names. Then, this dataset was refined using the following criteria: 1) fossil records and those lacking genus or species-level identification were removed; 2) introduced species were excluded; 3) duplicates were removed by considering locality, collector, and collection date; 4) Habitat fields were standardized by reclassifying terms such as 'dry forest', 'BTC', and '*selva baja caducifolia*' as SDTF, to extract

a list of SDTF-associated species (see S1 Table); 5) localities without coordinates were georeferenced, prioritizing species with few georeferenced records; and 6) species classification followed the criteria of the Angiosperm Phylogeny Group [48] and species names were validated using the Plants of the World Online (POWO; available at: https://powo.science.kew.org/). Besides, the resulting SDTF species list was further enriched with recently described taxa not yet included in POWO.

Species selection for the geographic delimitation of the SDTF biome was conducted in two steps. First, endemic species previously reported in the literature as occurring in Mexico were filtered and matched against the refined GBIF dataset to retrieve their occurrence records. These records were then spatially validated in ArcGIS 10.5 [49] by overlaying them on a map of Mexico to exclude georeferencing errors, retaining only records located within national boundaries. In a next step, species records were overlaid onto a previously defined SDTF polygon [4], and species were classified into three groups based on the proportion of their records within the biome, applying an approach similar to the Majority method [50]: (*i*) characteristic species, which include those with 50% or more of their records occurring within the SDTF polygon, representing the subregion where the majority of records are found; (*ii*) high-affinity species, comprising those with 75% or more of their records within the polygon; and (*iii*) strict endemics, referring to species whose records (100%) are confined exclusively within the SDTF boundaries. The 50% threshold used to define characteristic species ensures that each taxon is associated with a single dominant region [6]. The resulting species were subsequently used for the ecological niche modeling process.

### Environmental variables and species distribution modelling

A total of 58 environmental variables were initially considered at a spatial resolution of 30 arc-seconds (~1 km²). These included 26 climatic [51], nine edaphic, nine topographic, and 14 remote-sensing-derived variables [52]. To reduce multicollinearity, a Spearman correlation analysis was performed, selecting only the non-correlated variables (Spearman correlation < 0.7) that best explained the environmental variability by the studied species [53]. Additionally, to minimize spatial autocorrelation in the occurrence data, only localities separated by at least 5 km were retained, as environmental conditions exhibit significant variation at this scale [54]. As an accessible area (M) for all species [55], we utilized the World Wildlife Fund [56] database, selecting and merging those records that intersected with species occurrences.

Overall, SDMs were generated using the ENMTML package [57] in R 4.3.1 [58] within the RStudio environment [59]. Maxent was used as the modeling algorithm [60,61], with default settings implemented by ENMTML. Occurrence data were split according to sample size: species with 5–20 records were modeled using the bootstrap method, while those with ≥20 records were modeled using k-fold cross-validation [61], which identifies pixels with environmental suitability values equal to or greater than the lowest value at a known presence location [62]. Model performance was assessed using the Area Under the Curve (AUC), which ranges from 0 to 1 and is independent of the Lowest Presence Threshold (LPT). The LPT is a conservative criterion that classifies as suitable all pixels with environmental suitability values equal to or greater than the lowest value observed at a known presence location [62]. AUC values above 0.7 were considered indicative of good model performance. Models with AUC values below this threshold were discarded from further analyses.

### Delimitation of the Seasonally Dry Tropical Forest

The boundaries of Mexican SDTF were determined using a species-based modeling approach. First, individual species distribution models (SDMs) were generated for each of the three previously defined groups: strict endemics, high-affinity species, and characteristic species. These models were then stacked (i.e., summed) to obtain cumulative richness maps for each group. The stacking range for strict endemics included 1–13 species, for high-affinity species 1–65, and for characteristic species 1–122. To determine optimal thresholds for defining the SDTF based on stacked richness, we compared predicted distributions with the SDTF polygon proposed by Ortiz and Villaseñor [4], which was derived from Mexico's official vegetation map. For each group, we began by comparing areas predicted with a richness value of one species

and progressively increased the threshold (e.g., stacking ≥2 species, ≥ 3, and so on). For each threshold level, we calculated performance metrics using Cohen's Kappa, sensitivity, and specificity, as implemented in the *caret* package in R [63]. The final threshold was selected based on the highest Kappa value, ensuring a balance between high specificity and sensitivity values. Subsequently, the resulting SDTF polygon from each group (strict endemics, high-affinity species, and characteristic species) was then compared to previous proposals by DRYFLOR [36], Prieto-Torres et al. [31], and Ortiz & Villaseñor [4]. The comparison was done using a Kappa analysis in R 4.4.1 [58].

To identify which stacking group provided the most accurate delimitation of the SDTF, we used an independent validation dataset of 10,000 records of Asteraceae reported in herbarium labels for each of the five major biomes in Mexico (data provided by JLV, one of the authors). This plant family was selected because it has a wide distribution in the country and is well-represented in all vegetation types [64]. Records were spatially distributed across the five principal biomes of Mexico, and each point was overlaid with the SDTF polygons from the three proposals mentioned above. From this overlay, we generated a confusion matrix to calculate the number of true positives (TP), true negatives (TN), false positives (FP), and false negatives (FN). Based on these values, we calculated sensitivity and specificity, followed by Cohen's Kappa analysis to assess overall classification agreement. Additionally, we quantified underprediction (i.e., the proportion of SDTF records in areas where the evaluated proposal did not predict SDTF) and overprediction (i.e., the proportion of assumed absences in the expected presence zone) for each model proposal [65]. Finally, because the resulting models exhibited similar performance metrics, we applied a Wilcoxon rank-sum test to assess whether these differences were statistically significant.

## Results

From the STDF total records obtained from GBIF, 3,673 vascular plant species (441,774 records) were endemic to Mexico. For this pool, 228 species (12,411 records) contained a minimum number of records and were considered available for SDM (≥ 5 records). Their species models were executed using 34 uncorrelated environmental variables, and 96% of them demonstrated high predictive capacity, with ROC curve (AUC) values ranging from 0.70 to 1.00 (S2 Table). The final model stacking incorporated a total of 17 models of strict endemism species (i.e., STDF 100%) group (Fig 1A), 110 species in the high-affinity (i.e., STDF 75%) group (Fig 1B), and 223 species in the characteristic species (i.e., ≥ SDTF 50%) group (Fig 1C).

### Geographical boundaries of the SDTF in Mexico

The comparison of the three stacking models against the SDTF polygon proposed by Ortiz and Villaseñor [4] yielded the following results: the strict endemism model reached a maximum at the two-species range with a value of k = 0.412; the high-affinity SDTF species stacking model obtained the highest Kappa value at the three-species range with k = 0.52; and the characteristic species-based model reached the highest Kappa value at the 10-species range with k = 0.54 (see S3 Table). Based on the evaluation criteria—i.e., Kappa coefficient, sensitivity, specificity, and minimized underprediction and overprediction—the model composed of high-affinity species was selected as the most accurate representation of the SDTF distribution in Mexico (Table 1). Although the Kappa coefficient values of the high-affinity and characteristic species models were relatively close, a Wilcoxon rank-sum test revealed a statistically significant difference between them ($W = 48,895,000$, *p*-value = $8.36 \times 10^{-5}$), supporting the selection of the high-affinity model as the most robust.

The geographic extent of the SDTF, as predicted by the threshold stacking of the high-affinity species models, cover ca. 14% (278,581.9 km²) of Mexico's total land area (Fig 2). The distribution is mainly concentrated in the Pacific Coast (30%) and the Balsas Depression (19%) provinces, which also coincide with the areas of highest species richness (up to 65 species; Fig 2). In contrast, the SDTF occupies a much smaller area in the Gulf of Mexico province (0.7%), and the Yucatán Peninsula (0.03%).

When comparing the estimated areas between the proposals of SDTF in Mexico, the assembly proposed by DRYFLOR [36] and Prieto-Torres et al. [31] yielded similar cover values (24% and 23%, respectively). A low coverage value

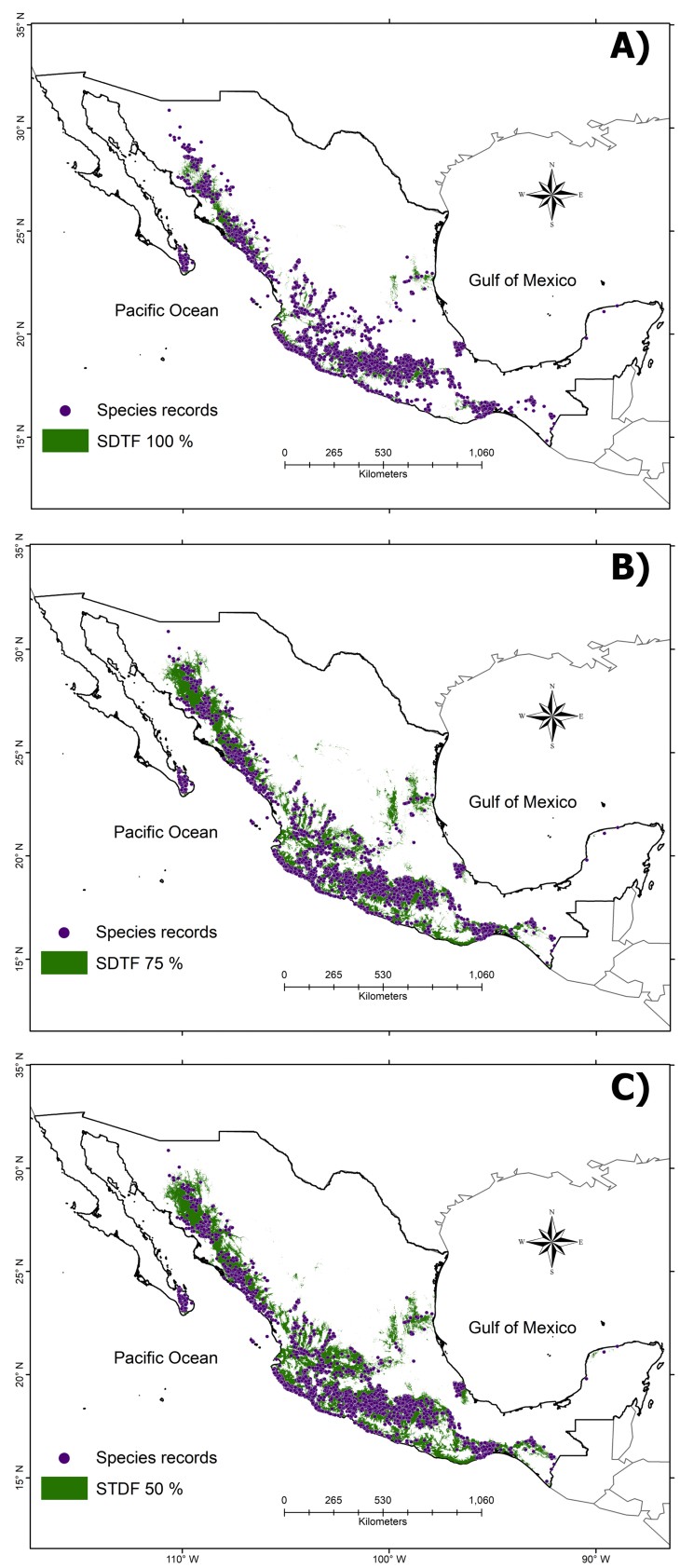

**Fig 1. Distribution of the Mexican Seasonally Dry Tropical Forest resulting from stacking individual models (green color) and the records of the 228 endemic species used to build up the ecological niche models (purple points).** The boundaries of the SDTF will be delineated by sorting the species into three groups, which differ in the percentage records of the species in this biome: A) strict endemic species (STDF 100%), B) high-affinity species (STDF 75%), and C) characteristic species (STDF 50%).

**Table 1. The evaluation metrics of the three stacking models for Mexican endemic species registered in the Seasonally Tropical Dry Forest (STDF), considering the collecting points of the Asteraceae species. Models are based on the percentage records of the species in this biome: strict endemic species (STDF 100%), high-affinity species (STDF 75%), and characteristic species (STDF 50%).**

| Models | Positive | | Negative | | Sensitivity | Specificity | Kappa | Prediction rate | |
| --- | --- | --- | --- | --- | --- | --- | --- | --- | --- |
| | True | False | True | False | | | | Under | Over |
| Strict endemism | 808 | 108 | 8075 | 1008 | 0.45 | 0.99 | 0.54 | 0.11 | 0.12 |
| High-affinity | 1284 | 569 | 7616 | 532 | 0.71 | 0.93 | 0.63 | 0.07 | 0.31 |
| Characteristic species | 1355 | 719 | 7465 | 461 | 0.75 | 0.91 | 0.62 | 0.06 | 0.35 |

(13%) was obtained with the proposal by Ortiz and Villaseñor [4]. This comparison reveals that discrepancies in SDTF predictions are more pronounced in northwestern Mexico, particularly in the piedmonts of the Sierra Madre Oriental, the Gulf of Mexico, and the Yucatán provinces (Fig 3). The comparison metrics for the SDTF proposals in Mexico provided relatively good Cohen's Kappa values. Specifically, our proposal strongly agrees with Prieto-Torres et al.'s values (0.58, p-value < 0.001). In contrast, a lower agreement was found when compared to the proposals by Ortiz and Villaseñor [4] (0.52, p-value < 0.001) and DRYFLOR [36] (0.40, p-value < 0.001) (Table 2).

The confusion matrices, using the Asteraceae occurrence records as independent field verification points, were applied to both the SDTF map developed in this study and the previously proposed delimitations, resulting in high values of sensitivity and specificity across all maps (Table 3). The SDTF map presented here showed a sensitivity of 0.71 and a specificity of 0.93, while the polygon proposed by Ortiz and Villaseñor [4] achieved the highest values for both metrics (sensitivity = 0.95; specificity = 0.98), indicating superior classification performance. Regarding Cohen's Kappa evaluation, which means the overall agreement between predicted and observed classifications, the Ortiz and Villaseñor proposal [4] again exhibited the highest accuracy (Kappa = 0.92, p-value < 0.001), followed by our proposal (Kappa = 0.63, p-value < 0.001) and that of Torres-Prieto et al. [31] (Kappa = 0.61, p-value < 0.001). The DRYFLOR [36] proposal obtained the lowest agreement values (Kappa = 0.5). On the other hand, in terms of underprediction and overprediction, the map by Ortiz and Villaseñor [4] again outperformed the others, with the lowest values (underprediction = 0.01; and overprediction = 0.08). In this regard, our model showed slightly higher values (underprediction = 0.06 and overprediction = 0.31), which is consistent with the findings of Prieto-Torres et al. [31] (underprediction = 0.03 and overprediction = 0.41). The DRYFLOR [36] polygon presented the highest error rates with 0.06 and 0.47 for underprediction and overprediction, respectively (Table 3).

## Discussion

Our study focused on defining a more accurate strategy for delineating the boundaries of biomes, using the Mexican SDTF as a case study. The floristic richness of Mexican SDTF is particularly notable, encompassing a total of 7,898 species—exceeding the 6,789 species previously reported by [66]—and exhibiting high rates of floristic turnover across its distribution [36,67–69]. This biome is also characterized by an exceptionally high level of endemism [36,39], with 3,673 species restricted to Mexico, suggesting diversification rates that likely surpass those observed in other tropical dry forest regions [36]. Remarkably, 449 of these species are strictly confined to SDTF areas, reflecting a high degree of specialization and adaptation to the local climatic conditions. All these factors highlight the outstanding conservation importance of this biome and its critical role in maintaining Mexico's biodiversity [66–68,70].

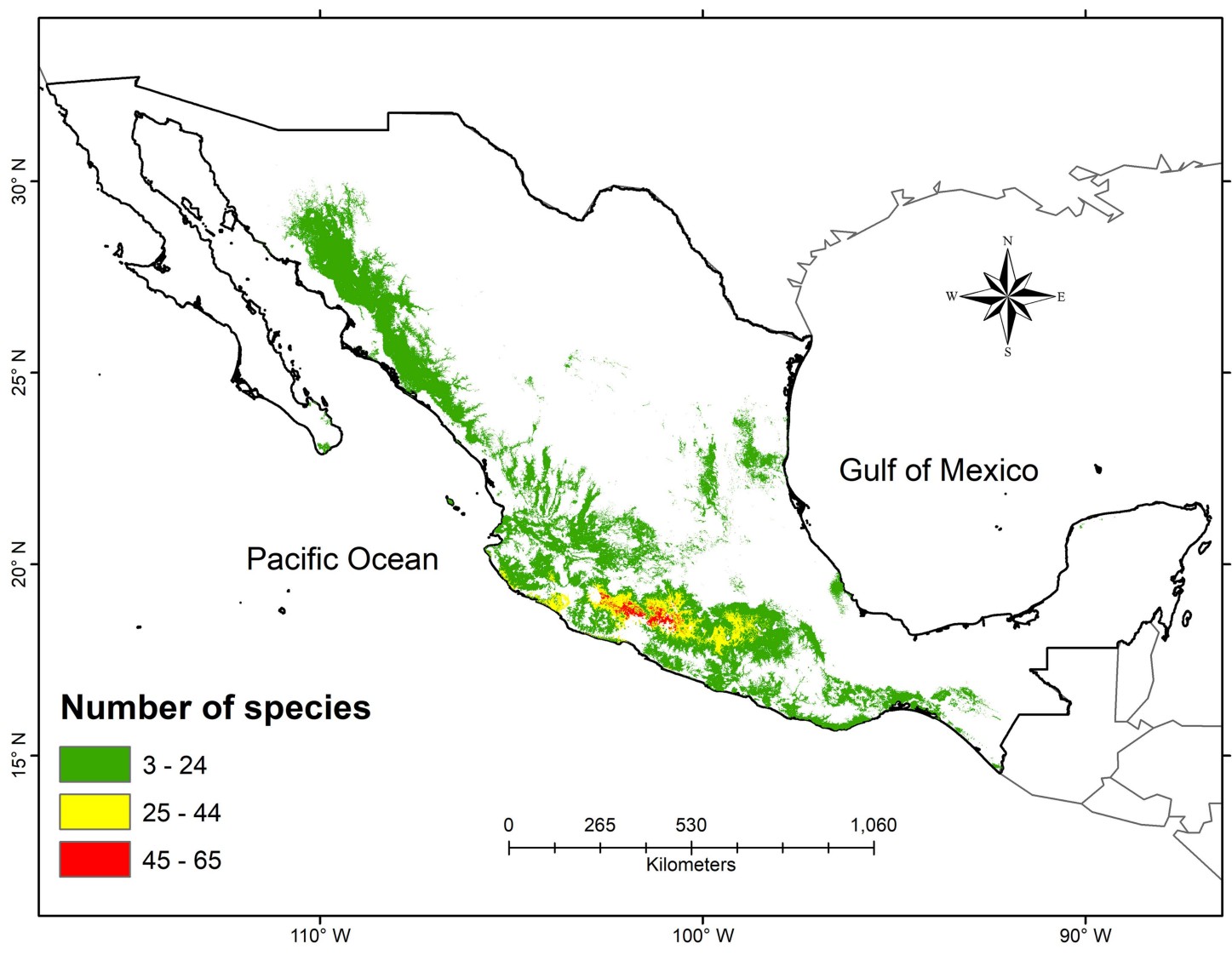

**Fig 2. Number of endemic species to the Seasonally Dry Tropical Forest in Mexico based on stacking distribution models of high-affinity species.** The colors represent the categories of model stacking, with the red indicating the area with the highest species richness.

Species selection plays a pivotal role in defining plant communities, as species exhibit differential responses to environmental gradients. In this context, models based on broadly distributed species may fail to capture the spatial patterns of endemic or characteristic taxa, even if they share specific environmental requirements [69]. In our study, the model based on species with high affinity to the biome statistically provided the most accurate delimitation of the biome's boundaries. Conversely, when only strictly endemic species were used, the general SDTF patterns were recovered; however, the model was overly restrictive, particularly failing to adequately represent the portion of the SDTF distributed across the Yucatán Peninsula. This is likely due to the limited presence of endemic or high-affinity species in that geographic area. Despite the role of ecological niche conservatism in shaping the distributional patterns of biota in Mesoamerican seasonally dry forests, several cases of niche divergence have been reported for the biota of the Yucatán Peninsula [71], potentially because of environmental niche loss. This may explain the main pattern observed in our study.

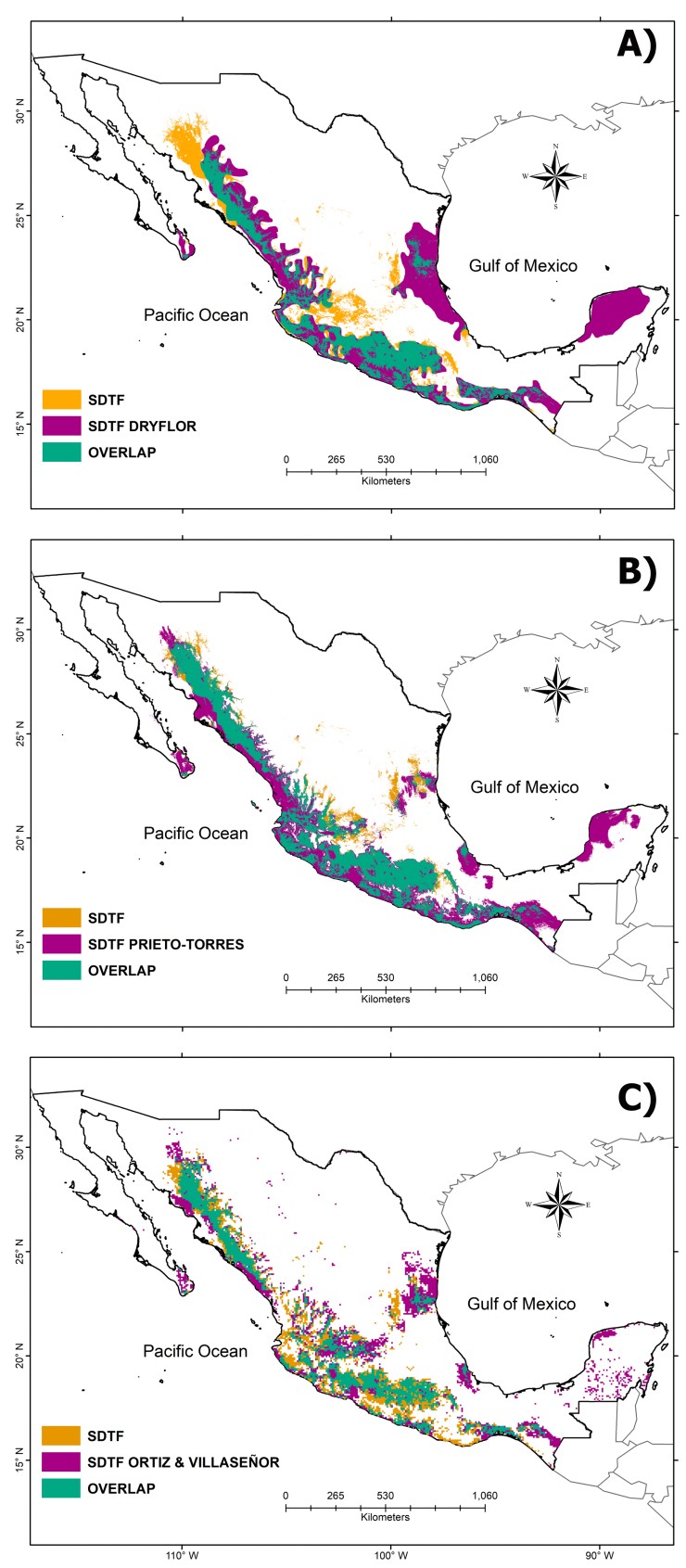

**Fig 3. Comparison of different proposals that delimit Mexico's Seasonally Dry Tropical Forest and this study. A)** DRYFLOR (2016), **B)** Prieto-Torres et al. (2016), and **C)** Ortiz and Villaseñor (2025).

**Table 2. Cohen's Kappa comparisons among the five proposals for the Seasonally Dry Tropical Forest in Mexico.**

| Proposals | 1 | 2 | 3 | 4 |
|---|---|---|---|---|
| 1. DRYFLOR (2016) | 1 | | | |
| 2. Prieto-Torres et al. (2016) | 0.58 | 1 | | |
| 3. Ortiz & Villaseñor (2025) | 0.40 | 0.51 | 1 | |
| 4. This proposal (high-affinity species) | 0.40 | 0.58 | 0.52 | 1 |

**Table 3. Evaluation metrics of the proposed Seasonally Dry Tropical Forest in Mexico, considering Asteraceae points.**

| Proposals | Positive | | Negative | | Sensitivity | Specificity | Kappa | Prediction rate | |
|---|---|---|---|---|---|---|---|---|---|
| | True | False | True | False | | | | Under | Over |
| 1 | 1362 | 1230 | 6954 | 454 | 0.75 | 0.85 | 0.51 | 0.06 | 0.47 |
| 2 | 1560 | 1108 | 7076 | 256 | 0.86 | 0.86 | 0.61 | 0.03 | 0.41 |
| 3 | 1721 | 152 | 8032 | 95 | 0.95 | 0.98 | 0.92 | 0.01 | 0.08 |
| 4 | 1284 | 569 | 7616 | 532 | 0.71 | 0.93 | 0.63 | 0.06 | 0.31 |

Previous community delimitation studies have often paid limited attention to rigorous species selection, focusing instead on statistical methodologies, thereby neglecting an evaluation of the strengths and limitations of different species selection strategies. Since s-SDM tends to underpredict and overpredict, it is essential to work more carefully on species selection and the data quality used in each species distribution model. This is important because even based on those characteristic species (i.e., with at least 50% of their records in the SDTF), we observed higher levels of overestimation in the prediction of biomes boundaries. In fact, when comparing our observed patterns of species richness with previous proposals, we found relatively low spatial agreement among sites of high species overlap. For instance, Prieto-Torres et al. [31] reported greater species overlap in the highlands of Chiapas and the southern Pacific Coast—areas that, in our analysis, exhibited lower overlap (< 32%). This discrepancy likely stems from the use of widely distributed species with broad ecological tolerances, such as *Cochlospermum vitifolium* (Willd.) Spreng. (Bixaceae), which is distributed from Mexico to Northern South America and occurs in four of the five major Mexican biomes recognized by Villaseñor & Ortiz [4]. In contrast, our results are more consistent with Trejo and Dirzo [67], who identified the Balsas Depression and the central part of the Pacific Coast as hotspots of species richness.

The inclusion of generalist species with broad ecological tolerances may obscure richness patterns in biomes characterized by high environmental and topographic heterogeneity [67,72]. Although the proposal by Manrique-Ascencio et al. [73] sought to represent broader community structure by incorporating species from five SDTF-associated plant families, their framework—similar to that of Prieto-Torres et al. [31]—relied heavily on widespread taxa that may extend beyond the biome's environmental limits. It is essential to note, however, that the primary objective of Manrique-Ascencio et al. [73] was not to define the boundaries of the SDTF, but rather to examine patterns of species richness within the biome. In contrast, our targeted selection of species with high ecological fidelity played a crucial role in revealing patterns of richness and community structure that more accurately represent the environmental and geographic characteristics of the SDTF.

Proposals that base their geographical delimitation strategy on expert knowledge—such as those proposed by Ortiz and Villaseñor [4] and DRYFLOR [36]—are challenging to compare due to the lack of supplementary data on diversity drivers (e.g., species richness), and their omission of the internal community structure [5,72]. This is a critical gap that

s-SDM positively consider and can address. Although both expert-based and model-based strategies can produce similar area estimates [74], expert-based maps often yield more conservative boundaries [73]. In the case of poorly known taxa, they may introduce spatial and taxonomic biases [75]. This would justify the lower area estimate in the proposal by Ortiz and Villaseñor [4].

The national forest inventory in Mexico has 26,220 sampling sites across the territory. However, only 41.8% provide usable data, resulting in incomplete knowledge of tree species distributions [76]. In contrast, s-SDMs offer spatially continuous and reproducible predictions based on available data [5]. By incorporating a diverse set of species exclusive to the SDTF and representing various growth forms [77], our study reduced omission and commission errors relative to other studies that focus on a single growth form or functional group [e.g., 5]. Consequently, our results could provide a more robust foundation for understanding macroecological patterns of this biome across the country [5,74,75].

Although s-SDMs, like macroecological models, tend to overpredict, especially in species-poor regions [17,78,79], they outperform inventory-based or multivariate methods when data on species distributions are limited [5,80]. In our case, the relatively low underprediction and overprediction values suggest a good classification performance. Ortiz and Villaseñor [4] obtained the lowest error values, likely due to their use of reclassified and interpolated vegetation data from INEGI Series VII [81]. In contrast, the expert-based DRYFLOR [36] map does not capture macroecological patterns as effectively as approaches based on species groups with broader distribution or functional groups [e.g., 31, 80].

Undoubtedly, paying attention to the quality of occurrence data and species selection in each model is essential for improving the reliability of s-SDMs outputs and achieving spatial predictions that more accurately reflect ecological processes and community structure in nature [82]. Our results demonstrate that endemism is a valuable surrogate for delimiting plant communities. Indeed, the patterns derived from endemic species align well with those from more comprehensive vegetation classifications, such as the INEGI assessment. Therefore, this methodological framework can also be applied to other vegetation types of ecological and conservation interest.

## Conclusions

The SDTF in Mexico represents a key biome for biodiversity conservation due to its exceptional species richness, high levels of endemism, and elevated levels of beta diversity. Our evaluation of SDMs revealed substantial variation in the estimated extent of the SDTF, underscoring significant discrepancies among previous delineations. These findings support the use of high-affinity endemic species as reliable surrogates for defining communities shaped by specific environmental conditions, such as the SDTF. Refining predictive modeling approaches is therefore essential for improving biome delineation accuracy, which in turn enhances the effectiveness of conservation planning and ensures the protection of areas with the most significant biodiversity value. Future research could expand this framework by incorporating additional components, such as beta-diversity metrics, phylogenetic turnover, and analyses based on the complete flora, to further assess the validity of endemism as a surrogate criterion for biome boundaries. Moreover, examining species turnover across transitional zones could help identify areas of ecological overlap with neighboring biomes that were not addressed in this study. Ultimately, this integrative approach can be extended to other biomes, contributing to more effective strategies for biodiversity conservation, ecosystem management, and ecological restoration at national and regional scales.

## Supporting information

**S1 Table. List of species present in the seasonally dry tropical forest in Mexico and the number of records obtained from the GBIF database.**
(DOCX)

**S2 Table. List of modeled species, modeling metrics, and model evaluation metrics.**
(XLSX)

**S3 Table. Threshold analysis for model stacking to delineate the boundaries of the SDTF in Mexico.**
(XLSX)

## Acknowledgments

The first author gratefully acknowledges financial support from the Postdoctoral Scholarship Program (DGAPA-UNAM). We thank CONABIO and the Instituto de Biología, UNAM, for allowing access to the information stored in the SNIB-REMIB and UNIBIO databases, respectively, which were a fundamental part of the analysis presented here.

## Author contributions

**Conceptualization:** Mayra Flores-Tolentino, José Luis Villaseñor, Guillermo Ibarra-Manríquez.

**Formal analysis:** Mayra Flores-Tolentino, José Luis Villaseñor, Enrique Ortíz, Guillermo Ibarra-Manríquez.

**Investigation:** Mayra Flores-Tolentino.

**Methodology:** Mayra Flores-Tolentino, José Luis Villaseñor, Enrique Ortíz, Guillermo Ibarra-Manríquez.

**Supervision:** David A. Prieto-Torres.

**Visualization:** Mayra Flores-Tolentino.

**Writing – original draft:** Mayra Flores-Tolentino.

**Writing – review & editing:** Mayra Flores-Tolentino, José Luis Villaseñor, Enrique Ortíz, David A. Prieto-Torres, Guillermo Ibarra-Manríquez.

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
