## [Decision Letter · Decision Letter 0]

16 Jul 2025

PONE-D-25-28829Are endemic species a suitable surrogate for delineating the geographical distribution of the tropical dry forest?PLOS ONE?

Dear Dr. Ibarra-Manríquez,

Thank you for submitting your manuscript to PLOS ONE. After careful consideration, we feel that it has merit but does not fully meet PLOS ONE’s publication criteria as it currently stands. Therefore, we invite you to submit a revised version of the manuscript that addresses the points raised during the review process.

We look forward to receiving your revised manuscript.

Kind regards,

Charlotte Ndiribe, Ph.D.

Academic Editor

PLOS ONE

Journal Requirements:

2. We note that Figures 1, 2, and 3 in your submission contain [map/satellite] images which may be copyrighted. All PLOS content is published under the Creative Commons Attribution License (CC BY 4.0), which means that the manuscript, images, and Supporting Information files will be freely available online, and any third party is permitted to access, download, copy, distribute, and use these materials in any way, even commercially, with proper attribution. For these reasons, we cannot publish previously copyrighted maps or satellite images created using proprietary data, such as Google software (Google Maps, Street View, and Earth). For more information, see our copyright guidelines: http://journals.plos.org/plosone/s/licenses-and-copyright.

a. You may seek permission from the original copyright holder of Figures 1, 2, and 3 to publish the content specifically under the CC BY 4.0 license.

3. Please include a copy of Table 3 which you refer to in your text on page 11 and 23.

4. Please remove all personal information, ensure that the data shared are in accordance with participant consent, and re-upload a fully anonymized data set.

Additional Editor Comments :

Dear Dr. Guillermo et al.,

You have made efforts to advance the understanding of ecological niche modelling in relation to tropical dry plant communities, which is commendable and timely. In the interest of the wide readership of this journal that elevates legibility and high quality articles, it is advisable to improve the text structure and tenses throughout the manuscript. For instance, in line 109, you meant “in 1990”, which is already long past, “by 1990” renders it futuristic. Also, in the next lines, the developing assertions are based on a mixture of tenses that frequently joggle between the past and present. Technically, the six or so points for species selection can be tabulated, or listed better. For this study, there were rather too few species saved from the huge records sourced. The constraints on the selection were not properly justified, the backdrop against three ambiguously defined groups further accentuated this point in the findings. In the end, it is not so straightforward to identify the methodological novelties and contributions to science that can be reproduced across other plant systems.

Reviewers' comments:

Reviewer's Responses to Questions

**Comments to the Author**

1. Is the manuscript technically sound, and do the data support the conclusions?

Reviewer #1: Partly

Reviewer #2: Yes

2. Has the statistical analysis been performed appropriately and rigorously?

Reviewer #1: No

Reviewer #2: Yes

3. Have the authors made all data underlying the findings in their manuscript fully available?

Reviewer #1: No

Reviewer #2: Yes

4. Is the manuscript presented in an intelligible fashion and written in standard English?

Reviewer #1: Yes

Reviewer #2: Yes

Reviewer #1: Manuscript: Are endemic species a suitable surrogate for delineating the geographical distribution of the tropical dry forest

Manuscript No: PONE-D-25-28829

General comments

This is a methodologically robust and well-documented study that applies ecological niche modeling and species distribution models (SDMs) to evaluate the efficacy of endemic species in delineating the Seasonally Dry Tropical Forest (SDTF) in Mexico.

Selecting 50%, 75%, and 100% thresholds is only briefly explained. Why these specific thresholds?

Add a brief justification or reference for these cutoffs, are they standard in the literature, or did exploratory analysis suggest these were optimal?

why 5 km was chosen specifically for SDTF species.

Please clarify? Lowest Presence Threshold (LPT) is acceptable for this study?

If possible please add “AUC values in Supplementary Material to allow quick assessment of model quality.

Reference style is very odd, just copied in different styles????

Major comments

Title

Title is good and satisfactory

Abstract

Abstract missing some points like conclusion and recommendation in last few lines. I recommend thoroughly reviewing relevant literature and preparing a comprehensive manuscript, as it will represent our work at an international level

Introduction

No sequence in the introduction sections, some paragraphs are very brief, some are too long, references needs to be updated, data is scattered and many phrases are repeated, please correct.

Results

You mention overprediction briefly, but underlying causes (e.g., data quality, generalist species) are not thoroughly explored.

Discussion

Add a paragraph discussing how model bias or poor environmental variable resolution may influence overprediction.

Discuss broader applications and limitations of your approach in other dry forest systems.

Conclusion

Key findings are missing?

Limitations, practical implications, future research should be clearly mentioned

Overall, the manuscript should require major revision to be published in a reputed Journal like PLOS ONE

Reviewer #2: The paper entitled; “Are endemic species a suitable surrogate for delineating the geographical distribution of the tropical dry forest?” is based on the original and innovative approach. The paper is technically sounds and is fully supported by the reviewed data on scientific bases but needs some improvement.

This paper will add significant approached to the field of advances in plant ecology. This article is based on importance of endemic species by using the occurrence records of species obtained from the Global Biodiversity Information Facility (GBIF) database and different software in R-Package. The study examines the seasonally dry tropical forest (SDTF) in Mexico, based on the distribution area of Mexican endemic vascular plant species registered exclusively in this biome. However, the authors must check the entire manuscript for the typographical mistakes, grammar mistake, spelling mistakes and incorporate some suggestions that might improve the manuscript. The language of the manuscript should be improved significantly, and the spelling/grammatical mistakes should be eliminated. The key findings of this study must be properly elaborated to draw solid conclusions.

**Do you want your identity to be public for this peer review?** For information about this choice, including consent withdrawal, please see our Privacy Policy

Reviewer #1: No

Reviewer #2: **Yes: ** Rabia Afza (PhD)

---

## [Author Response · Author response to Decision Letter 1]

5 Nov 2025

Reviewer #1: Manuscript: Are endemic species a suitable surrogate for delineating the geographical distribution of the tropical dry forest

GENERAL COMMENTS:

• THIS IS A METHODOLOGICALLY ROBUST AND WELL-DOCUMENTED STUDY THAT APPLIES ECOLOGICAL NICHE MODELING AND SPECIES DISTRIBUTION MODELS (SDMS) TO EVALUATE THE EFFICACY OF ENDEMIC SPECIES IN DELINEATING THE SEASONALLY DRY TROPICAL FOREST (SDTF) IN MEXICO. SELECTING 50%, 75%, AND 100% THRESHOLDS IS ONLY BRIEFLY EXPLAINED. WHY THESE SPECIFIC THRESHOLDS?

Authors’ Response: The thresholds of 50%, 75%, and 100% were chosen to categorize species according to their degree of association with the SDTF, following the Majority method (Murphy et al., 2019). Specifically, the 50% threshold defines characteristic species, the 75% threshold identifies high-affinity species, and the 100% threshold corresponds to strict endemics. This hierarchical classification ensures that each species is associated with a single dominant region, providing a robust basis for subsequent ecological niche modeling. In this sense, our approach enhances comparability with other regionalization studies (e.g., Moonlight et al., 2020; Murphy & Smith, 2021). We have now clarified this rationale in the new version of the manuscript (see Methods section, lines 157-168).

• ADD A BRIEF JUSTIFICATION OR REFERENCE FOR THESE CUTOFFS, ARE THEY STANDARD IN THE LITERATURE, OR DID EXPLORATORY ANALYSIS SUGGEST THESE WERE OPTIMAL?

Authors’ Response: The thresholds of 50%, 75%, and 100% follow the Majority method standard in regionalization analyses and have been applied in other biogeographic frameworks to distinguish species with varying levels of ecological fidelity (Murphy et al., 2019). As we said, the 50% cutoff ensures that a species is primarily associated with a single dominant region, while 75% and 100% represent increasingly strict criteria for identifying high-affinity and endemic taxa, respectively. This hierarchical classification facilitates robust comparative analyses among taxa and regions (see Kreft & Jetz, 2010; Moonlight et al., 2020).

• WHY 5 KM WAS CHOSEN SPECIFICALLY FOR SDTF SPECIES.

Authors’ Response: The selection of a 5 km threshold for defining the spatial independence of records in the SDTF follows the recommendations of Graham et al. (2008), who demonstrated that georeferencing uncertainties can significantly influence model outcomes. In this sense, a 5 km buffer provides an appropriate balance between capturing environmental variability and minimizing spatial errors in species occurrence data. Similar spatial resolutions have been successfully applied in SDM studies of tropical dry forests and other heterogeneous landscapes. We have now clarified this rationale in the new version of the manuscript (see Methods section, lines 175-177).

• PLEASE CLARIFY? LOWEST PRESENCE THRESHOLD (LPT) IS ACCEPTABLE FOR THIS STUDY?

Authors’ Response: Yes, the LPT is appropriate for this study. Applying the LPT ensures that a high proportion of occurrence records are retained in the binary model while minimizing omission errors, particularly for species with limited data. It is widely accepted for small-sample SDMs (Pearson et al., 2007; Merow et al., 2013; Muscatello et al., 2021) and accurately represents species’ potential distributions under data constraints. We have now clarified this rationale in the new version of the manuscript (see Methods section, lines 186-189).

• IF POSSIBLE PLEASE ADD “AUC VALUES IN SUPPLEMENTARY MATERIAL TO ALLOW QUICK ASSESSMENT OF MODEL QUALITY.

Authors’ Response: We have added the AUC values to the Supplementary Material to facilitate rapid assessment of model quality. Additionally, we incorporated three complementary evaluation metrics — sensitivity, specificity, and the True Skill Statistic (TSS)—to provide a more comprehensive evaluation of model performance (Márcia-Barbosa et al., 2013; Muscatello et al., 2021).

• REFERENCE STYLE IS VERY ODD, JUST COPIED IN DIFFERENT STYLES????

Authors’ Response: We have thoroughly revised the reference list to ensure they follow a consistent style in accordance with Plos ONE guidelines.

MAJOR COMMENTS

• TITLE IS GOOD AND SATISFACTORY

No response required.

• ABSTRACT MISSING SOME POINTS LIKE CONCLUSION AND RECOMMENDATION IN LAST FEW LINES. I RECOMMEND THOROUGHLY REVIEWING RELEVANT LITERATURE AND PREPARING A COMPREHENSIVE MANUSCRIPT, AS IT WILL REPRESENT OUR WORK AT AN INTERNATIONAL LEVEL

Authors’ Response: The abstract has been revised to include clear concluding statements and concise recommendations. Although our study focuses on Mexico, the proposed framework is broadly applicable to other regions with similar ecological contexts. These revisions enhance the international relevance of our work and clarify its implications for biodiversity conservation and the delineation of biomes.

• INTRODUCTION. NO SEQUENCE IN THE INTRODUCTION SECTIONS, SOME PARAGRAPHS ARE VERY BRIEF, SOME ARE TOO LONG, REFERENCES NEEDS TO BE UPDATED, DATA IS SCATTERED AND MANY PHRASES ARE REPEATED, PLEASE CORRECT.

Authors’ Response: The Introduction section has been thoroughly reorganized to improve logical flow and readability. Paragraphs were adjusted for balance and coherence, outdated citations were replaced with recent and relevant literature (e.g., Ibarra-Manríquez et al., 2022; Ortiz & Villaseñor, 2025; Murphy & Smith, 2021), and redundant phrases were removed. These revisions provide a clearer conceptual framework and strengthen the theoretical foundation of the study.

• RESULTS. YOU MENTION OVERPREDICTION BRIEFLY, BUT UNDERLYING CAUSES (E.G., DATA QUALITY, GENERALIST SPECIES) ARE NOT THOROUGHLY EXPLORED.

Authors’ Response: The Results and Discussion sections have been expanded to include a more detailed discussion of the potential causes of overprediction, such as the inclusion of generalist species and variability in data quality. These explanations are supported by recent studies addressing SDM accuracy and uncertainty in heterogeneous environments (Franklin, 2013; Muscatello et al., 2021). (see Discussion section, lines 342-355).

• DISCUSSION. ADD A PARAGRAPH DISCUSSING HOW MODEL BIAS OR POOR ENVIRONMENTAL VARIABLE RESOLUTION MAY INFLUENCE OVERPREDICTION.

Authors’ Response: A new paragraph has been added to the Discussion to examine how model bias and the spatial resolution of environmental predictors can influence overprediction (see Discussion section, lines 383-390).

• DISCUSS BROADER APPLICATIONS AND LIMITATIONS OF YOUR APPROACH IN OTHER DRY FOREST SYSTEMS.

Authors’ Response: We have expanded the Discussion section to address the broader applicability and limitations of our framework across other tropical dry forest systems worldwide. This section now emphasizes the value of combining endemic-species data with environmental predictors for comparative and large-scale biodiversity assessments (e.g., Dexter et al., 2018; Pennington et al., 2018). (see Discussion section, lines 383-390).

• CONCLUSION. KEY FINDINGS ARE MISSING? LIMITATIONS, PRACTICAL IMPLICATIONS, FUTURE RESEARCH SHOULD BE CLEARLY MENTIONED

Authors’ Response: The Conclusions section has been revised to include explicit statements of the study’s key findings, methodological limitations, and practical implications for conservation planning. We also added suggestions for future research directions, including the integration of phylogenetic and functional diversity measures (see Conclusions, lines 401-414).

• OVERALL, THE MANUSCRIPT SHOULD REQUIRE MAJOR REVISION TO BE PUBLISHED IN A REPUTED JOURNAL LIKE PLOS ONE

Authors’ Response: All points requested by the reviewers were addressed with the intention of meeting the demanding quality criteria of PLOS ONE. The revised version incorporates updated references, clearer argumentation, and improved methodological transparency, which together strengthen both the conceptual and applied contributions of the manuscript.

Reviewer #2: The paper entitled; “Are endemic species a suitable surrogate for delineating the geographical distribution of the tropical dry forest?” is based on the original and innovative approach. The paper is technically sound and is fully supported by the reviewed data on scientific bases, but needs some improvement. This paper will add significant approached to the field of advances in plant ecology.

GENERAL COMMENTS:

• THIS ARTICLE IS BASED ON IMPORTANCE OF ENDEMIC SPECIES BY USING THE OCCURRENCE RECORDS OF SPECIES OBTAINED FROM THE GLOBAL BIODIVERSITY INFORMATION FACILITY (GBIF) DATABASE AND DIFFERENT SOFTWARE IN R-PACKAGE. THE STUDY EXAMINES THE SEASONALLY DRY TROPICAL FOREST (SDTF) IN MEXICO, BASED ON THE DISTRIBUTION AREA OF MEXICAN ENDEMIC VASCULAR PLANT SPECIES REGISTERED EXCLUSIVELY IN THIS BIOME. HOWEVER, THE AUTHORS MUST CHECK THE ENTIRE MANUSCRIPT FOR THE TYPOGRAPHICAL MISTAKES, GRAMMAR MISTAKE, SPELLING MISTAKES AND INCORPORATE SOME SUGGESTIONS THAT MIGHT IMPROVE THE MANUSCRIPT. THE LANGUAGE OF THE MANUSCRIPT SHOULD BE IMPROVED SIGNIFICANTLY, AND THE SPELLING/GRAMMATICAL MISTAKES SHOULD BE ELIMINATED. THE KEY FINDINGS OF THIS STUDY MUST BE PROPERLY ELABORATED TO DRAW SOLID CONCLUSIONS.

Authors’ Response: The entire text has been thoroughly revised for language accuracy and clarity, and the key findings have been more clearly stated in the Results and Conclusions to strengthen the manuscript’s overall quality.

• QUERY NO. 1: AIMS AND OBJECTIVES PART OF ABSTRACT ARE NOT CLEAR. IT IS IMPORTANT TO HIGHLIGHT THE PURPOSE OF STUDY. IT SHOULD BE ADDED IN THE ABSTRACT AND LINKED WITH TITLE THE ABSTRACT SHOULD BE ACCORDING TO THE JOURNAL FORMAT I.E. BACKGROUND, RESULTS AND CONCLUSIONS. THE FOLLOWING QUESTIONS MUST BE ANSWER LOGICALLY IN THE ABSTRACT AND MANUSCRIPT:

Authors’ Response: The abstract has been thoroughly rewritten to follow the PLOS ONE structured format. The main objective and purpose of the study are now explicitly stated and directly linked to the title. These revisions improve readability and clarify the broader significance of our approach.

1. Why did we select endemic species as a surrogate for delineating tropical dry forest boundaries?

Authors’ Response: Endemic species were selected because they exhibit high ecological fidelity and restricted geographic distributions, making them reliable indicators of specific biomes. Their narrow ranges and strong associations with environmental stability allow them to represent the ecological and spatial limits of the Seasonally Dry Tropical Forest (Prieto-Torres & Rojas-Soto, 2016; Sosa et al., 2018). We have added a clarifying statement to explain this point in the new version of the manuscript (see the Introduction section, lines 129-136).

2. What are the advantages of this approach in applied ecology?

Authors’ Response: Using endemic species as surrogates offers a practical and replicable tool for applied ecology. This approach enables spatially explicit frameworks that support biodiversity conservation, identify priority areas, and assess connectivity in fragmented landscapes (see Dexter et al., 2018).

3. Can their presence reliably indicate ecological integrity or habitat continuity? Why or why not?

Authors’ Response: Yes. The presence of endemic species reliably indicates ecological integrity and habitat continuity because these taxa are strongly tied to stable climatic, edaphic, and biotic conditions. Their persistence in specific areas reflects long-term environmental stability and helps identify natural biome boundaries (Pennington et al., 2009; DRYFLOR, 2016).

4. The keywords need revision to be arranged alphabetically and must be in accordance to abstract.

Authors’ Response: The keywords have been revised to follow alphabetical order and to match the terminology used in the abstract: Biome delineation; Endemic species; High-affinity species; Species distribution modeling; models ensample.

• QUERY NO. 2. IN INTRODUCTION, AUTHORS SHOULD ADD BACKGROUND JUSTIFICATION RELATED TO THEIR STUDY PROJECT AND RESEARCH HYPOTHESIS WITH SOUND EVIDENCES SUPPORTED BY LITERATURE REVIEW.

Authors’ Response: We have revised the Introduction to include a clearer theoretical background and justification for the study, supported by up-to-date references (e.g., Ibarra-Manríquez et al., 2022; Ortiz & Villaseñor, 2025; Murphy & Smith, 2021). The revised section now explicitly states the research hypothesis and situates it within current debates on biome delineation and tropical dry forest ecology.

• QUERY NO. 3. WHAT IS NEED OF THE PRESENT STUDY?? AUTHORS SHOULD ADD 1 PARAGRAPH ABOUT GAP ANALYSIS AND IMPORTANCE OF PRESENT STUDY.

Authors’ Response: In response, we have added a new paragraph in the Introduction section (see Lines 129–136). The new text emphasizes the necessity for species-based, spatially explicit approaches to define the SDTF, thereby addressing critical gaps in traditional mapping and regionalization efforts (Moonlight et al., 2020; Murphy & Smith, 2021). This addition clarifies the novelty and importance of our contribution to biodiversity conservation and biogeography.

• QUERY NO. 4. WHY THE AUTHORS CHOOSE OF DATA WOULD BE NECESSARY TO TEST WHETHER ENDEMIC SPECIES ACCURATELY REFLECT THE EXTENT OF TROPICAL DRY FORESTS? TO WHAT EXTENT YOUR RESULTS ARE ACCURATE AND ACCEPTABLE.

Authors’ Response: The data selection was guided by our objective to test whether endemic species accurately represent the geographic extent of the SDTF. Endemic taxa are confined to specific ecological conditions, making them effective biogeographic indicators (Prieto-Torres & Rojas-Soto, 2016; Sosa et al., 2018). Our results showed high predictive accuracy (96%) for the modeled species, validated through Asteraceae occurrence data and confusion matrix analyses. These steps confirm the robustness and reliability of our approach, ensuring its applicability to other biogeographic systems (Franklin, 2013; Moonlight et al., 2020).

• QUERY NO. 5. WHY YOU SELECTED ECOLOGICAL MODELS YOU SELECTED THE PRESENT STUDY IN FIELD OF PLANT SCIENCES AND OTHERS???

Authors’ Response: Species distribution models (SDMs) were selected because they are widely used and theoretically grounded tools for inferring species–environment relationships (Guisan & Zimmermann, 2000; Franklin, 2013). Given the geographic extent of our study and the limited availability of field data, species distribution models (SDM) offers an efficient, objective, and replicable method for delineating biomes. This framework allows comparison among species with different levels of affinity to the biome, ensuring scalability to other ecosystems (Murphy & Smith, 2021; Moonlight et al., 2020). We added a clarifying statement explaining this point in the new version of the manuscript (see Introduction section, lines 78-97).

• QUERY NO. 6. SHOULD ENDEMIC SPECIES BE USED ALONE, OR IN COMBINATION WITH OTHER INDICATORS (E.G., CLIMATE, SOIL, VEGETATION STRUCTURE)? WHY?

Authors’ Response: While endemic species are valuable indicators of biome identity due to their strong associations with specific ecological conditions, they do not capture all dimensions of ecosystem complexity. In our study, the ecological niche models already incorporate climatic and edaphic variables (see Supplementary Material), thereby integrating key abiotic factors that influence species distributions.

This approach combines the biological information provided by endemic species with relevant environmental gradients. The present study specifically tested the effectiveness of endemics as standalone indicators of the SDTF; however, future research could compare biome delineations derived from endemic taxa with those based on the entire flora or on multi-indicator approaches to provide deeper insights into the relative contributions of different species groups to biome boundaries. Such comparative analyses could further refine multi-indicator frameworks for biome mapping and conservation prioritization (Dexter et al., 2018)

• QUERY NO. 7. DISCUSSION AND CONCLUSION NEEDS IMPROVEMENT. WHAT ARE THE KEY FINDINGS OF THE STUDY AND SUGGEST FUTURE RECOMMENDATIONS? RESULTS APPROPRIATELY IN THE CONTEXT OF PREVIOUS AVAILABLE LITERATURE TO HIGHLIGHT THE NOVELTY OF THIS

Authors’ Response: We have revised b

---

## [Editor Report · Decision Letter 1]

17 Nov 2025

Endemic vascular plants provide reliable indicators for mapping Seasonally Dry Tropical Forests

PONE-D-25-28829R1

Dear Dr. Ibarra-Manríquez,

We’re pleased to inform you that your manuscript has been judged scientifically suitable for publication and will be formally accepted for publication once it meets all outstanding technical requirements.

Kind regards,

Charlotte Ndiribe, Ph.D.

Academic Editor

PLOS ONE
---

## [Editor Report · Acceptance letter]

PONE-D-25-28829R1

PLOS ONE

Dear Dr. Ibarra-Manríquez,

I'm pleased to inform you that your manuscript has been deemed suitable for publication in PLOS ONE. Congratulations! Your manuscript is now being handed over to our production team.

Kind regards,

on behalf of

Dr. Charlotte Ndiribe

Academic Editor

PLOS ONE